# Precise coordination between nutrient transporters ensures fertility in the malaria mosquito *Anopheles gambiae*

Iryna Stryapunina[1], Maurice A. Itoe[1], Queenie Trinh[1], Charles Vidoudez[2], Esrah Du[1], Lydia Mendoza[1,¤], Oleksandr Hulai[1], Jamie Kauffman[1], John Carew[1], W. Robert Shaw[1,3], Flaminia Catteruccia[1,3]*

1 Harvard T.H. Chan School of Public Health, Boston, Massachusetts, United States of America, 2 Harvard Center for Mass Spectrometry, Cambridge, Massachusetts, United States of America, 3 Howard Hughes Medical Institute, Chevy Chase, Maryland, United States of America

¤ Current address: University of Pennsylvania, Philadelphia, Pennsylvania, United States of America
* fcatter@hsph.harvard.edu

## Abstract

Females from many mosquito species feed on blood to acquire nutrients for egg development. The oogenetic cycle has been characterized in the arboviral vector *Aedes aegypti*, where after a bloodmeal, the lipid transporter lipophorin (Lp) shuttles lipids from the midgut and fat body to the ovaries, and a yolk precursor protein, vitellogenin (Vg), is deposited into the oocyte by receptor-mediated endocytosis. Our understanding of how the roles of these two nutrient transporters are mutually coordinated is however limited in this and other mosquito species. Here, we demonstrate that in the malaria mosquito *Anopheles gambiae*, Lp and Vg are reciprocally regulated in a timely manner to optimize egg development and ensure fertility. Defective lipid transport via *Lp* knockdown triggers abortive ovarian follicle development, leading to misregulation of Vg and aberrant yolk granules. Conversely, depletion of Vg causes an upregulation of *Lp* in the fat body in a manner that appears to be at least partially dependent on target of rapamycin (TOR) signaling, resulting in excess lipid accumulation in the developing follicles. Embryos deposited by Vg-depleted mothers are completely inviable, and are arrested early during development, likely due to severely reduced amino acid levels and protein synthesis. Our findings demonstrate that the mutual regulation of these two nutrient transporters is essential to safeguard fertility by ensuring correct nutrient balance in the developing oocyte, and validate Vg and Lp as two potential candidates for mosquito control.

## Author summary

Female mosquitoes bite humans to acquire nutrients for egg development. The shuttling of nutrients from the gut (where the blood is digested) to the ovaries (where eggs are produced) relies on several nutrient transporters to move through the mosquito circulation, including Lipophorin (Lp), a fat transporter expressed early and Vitellogenin (Vg), a large

**Data Availability Statement:** The numerical data are provided in the Supporting Information and are accessible from the Harvard Dataverse online

repository using the link https://doi.org/10.7910/DVN/C5UOZS.

**Funding:** F.C. is funded by the Howard Hughes Medical Institute (HHMI) as an HHMI investigator (www.hhmi.org), and by the National Institutes of Health (NIH) (R01AI148646, R01AI153404, www.nih.gov). I.S. is funded by Natural Sciences and Engineering Research Council of Canada (NSERC, www.nserc-crsng.gc.ca) as a postgraduate scholarship recipient (PGSD3 - 545866 - 2020). M.I. is funded by Charles A. King Trust postdoctoral research fellowship in basic science from Health Resources in Action (HRiA, www.hria.org). The findings and conclusions within this publication are those of the authors and do not necessarily reflect positions or policies of the HHMI or the NIH. The funders had no role in the study design, in data collection, analysis or interpretation, in the decision to publish, or the preparation of the manuscript.

**Competing interests:** The authors have declared that no competing interests exist.

protein expressed later that transports amino acids. We know relatively little of how these two nutrient transporters are successfully coordinated in any mosquito species, so we undertook to examine their interplay in the *Anopheles* malaria mosquito. We found that without Lp expression, Vg is incorrectly distributed within ovarian follicles and egg production is aborted, whereas without Vg, fat transport by Lp is not switched off in a timely manner. This results in excess fat and minimal protein deposition in eggs, rendering females completely infertile. We also find evidence that the mutual regulation of these transporters may be mediated by TOR signaling. As well as providing further insight into the regulation of essential reproductive processes, these results may aid in the development of malaria control strategies that aim to reduce the size of mosquito populations.

## Introduction

The *Anopheles gambiae* mosquito is one of the most important vectors for the transmission of *Plasmodium falciparum*, a malaria parasite that causes remarkable morbidity and mortality in sub-Saharan Africa and other tropical and subtropical regions [1]. Transmission starts when a female mosquito takes a blood meal from a human infected with the sexual stages of *Plasmodium*. At the same time as parasite development begins in the midgut, females use blood nutrients to start their reproductive cycle, which culminates in the development of a full set of eggs in about 2–3 days. The signalling cascades triggered by blood feeding and leading to successful egg development have been largely elucidated in *Aedes aegypti* mosquitoes. In this species, the ovarian ecdysteroidogenic hormone (OEH) and insulin-like peptides (ILPs) are released from the brain upon blood feeding, stimulating the production of the ecdysteroid ecdysone (E, which is synthetized from cholesterol) by the ovarian epithelium [2,3]. After transport to the fat body, E is converted to 20-hydroxyecdysone (20E, the active form of this steroid hormone), which binds to its nuclear receptor to trigger transcriptional cascades leading to the activation and repression of hundreds of genes. Among these genes is *Vitellogenin* (*Vg*), the main egg yolk protein precursor in oviparous species, which is transcribed in the fat body, peaking in production at 24 hours (h) post blood meal (PBM) [4]. After translation, Vg is then released from the fat body into the hemolymph, from where it is taken up by the ovaries by receptor-mediated endocytosis [5,6]. In the oocytes Vg is crystalized into vitellin, which forms the yolk bodies that the embryo uses as a nutritional source of amino acids [7–9].

Prior to *Vg* expression, the lipid transporter lipophorin (Lp) shuttles cholesterol and neutral lipids (mostly triglycerides [10]) from the midgut to the ovaries, starting the early phase of egg development [11,12]. It is unclear how *Lp* expression is regulated, although *ex vivo* experiments in *Ae. aegypti* have shown this lipid transporter to be upregulated upon fat body exposure to 20E [11]. After egg development is completed, if the female is mated, she will oviposit her eggs and return to the pre-blood meal metabolic state. At this point she is ready to begin another gonotrophic cycle, consisting of blood feeding, oogenesis and oviposition.

Besides triggering the synthesis of 20E through cholesterol uptake and E release by the ovaries, blood meal digestion and the subsequent influx of amino acids and ILPs results in the activation of the target of rapamycin (TOR) signalling pathway [4,13]. The integration of these nutritional signals leads to a TOR-mediated global regulation of translation and transcription of specific genes that control growth and metabolism [14], including *Vg* transcription in *Ae. aegypti* mosquitoes [15], reviewed in [4,13]. TOR regulates translation by directly phosphorylating S6 kinase (S6K) [16], which in turn phosphorylates S6, thus regulating translation of ribosomal proteins and translation elongation factors (reviewed in [4,13]). S6K also activates

the translation of AaGATAa, which promotes *Vg* transcription in *Ae. aegypti* [17]. Due to its central role in the integration of nutritional signals post blood meal, abrogation of TOR signalling results in reduced fecundity in *Ae. aegypti* and *Anopheles stephensi* [15,16,18,19].

Although successful oogenesis in mosquitoes is likely to be tightly dependent on the coordinated function of Lp and Vg, very limited information is available concerning whether and how these nutrient transporters are mutually regulated to ensure egg development and fertility. A study conducted in *An. gambiae* showed that *Lp* knockdown results in reduced expression of *Vg* after feeding on mice infected with the rodent malaria parasite *Plasmodium berghei*, while *Vg* depletion did not affect *Lp* expression [20]. No further studies have clarified the possible co-regulation of these factors in ensuring accurate nutrient deposition during oogenesis. Additionally, data concerning the mechanisms regulating egg development in *An. gambiae* are sparce despite the key importance of this species for malaria transmission. In these mosquitoes, knockdown of *Lp* (whose expression peaks at 12–18h PBM) has been shown to severely hamper egg development [20–22], revealing a similar role to *Aedes*. Interestingly, *Lp* levels were upregulated upon inhibition of 20E signalling via depletion of the nuclear EcR receptor, suggesting that in certain conditions 20E may repress its expression [22]. The role of Vg in *An. gambiae* reproduction has been studied even more marginally, with a single study reporting that its depletion results in fewer females developing mature oocytes [20].

Here we show that the functions of Lp and Vg are tightly linked in *An. gambiae*. Using functional knockdowns of these genes combined with electron microscopy, multi-omics and biochemistry analyses, we show that depletion of these factors results in profoundly deleterious effects on fecundity and fertility. While Lp is required for successful egg development, Vg is essential for fertility as its depletion leads to an early block in embryonic development. We also prove that the functions of these factors are mutually co-regulated, as Lp is needed for the correct incorporation of Vg in the developing oocytes while Vg's utilization of amino acids is required to terminate the Lp-mediated accumulation of lipids in the ovaries. Intriguingly, our data suggest that both induction of *Vg* after blood feeding and its downstream effects on Lp are mediated by TOR signalling, which in turn appears to be repressed by Vg following a blood meal. Our data reveal an intricate reproductive system based on the timely and mutually coordinated function of these nutrient transporters, and identify novel potential targets to interfere with the fertility of these important malaria vectors.

## Results

### *Lp* knockdown significantly impairs oogenesis and affects Vg localization

Previous studies have shown that *Lp* knockdown results in severely reduced oogenesis in *An. gambiae* females [20–22] but these studies did not characterize oocyte development over time and did not assess effects on fertility. We confirmed these findings by injecting double stranded RNA targeting *Lp* (ds*Lp)* prior to blood feeding (S1A Fig), which dramatically reduced the median number of eggs developed by females compared to controls (injected with ds*LacZ*) (Fig 1A). This reduction in egg numbers was characterized by a striking decrease in triglycerides in the ovaries at different time points post blood meal (PBM), paralleled by a substantial accumulation of these lipids in the midgut and fat body, consistent with a role of Lp in shuttling these lipids to the developing oocytes (Fig 1B).

We next examined how *Lp* knockdown affects levels of *Vg*, the major nutrient transporter, which, as mentioned above, is transcribed and translated in the fat body and incorporated into developing oocytes. Interestingly, *Lp*-knockdown females had aberrant Vg production and distribution. Not only was *Vg* expressed at lower levels upon Lp depletion (S1B Fig), consistent with previous work [20], but also this yolk protein showed atypical localization. While in

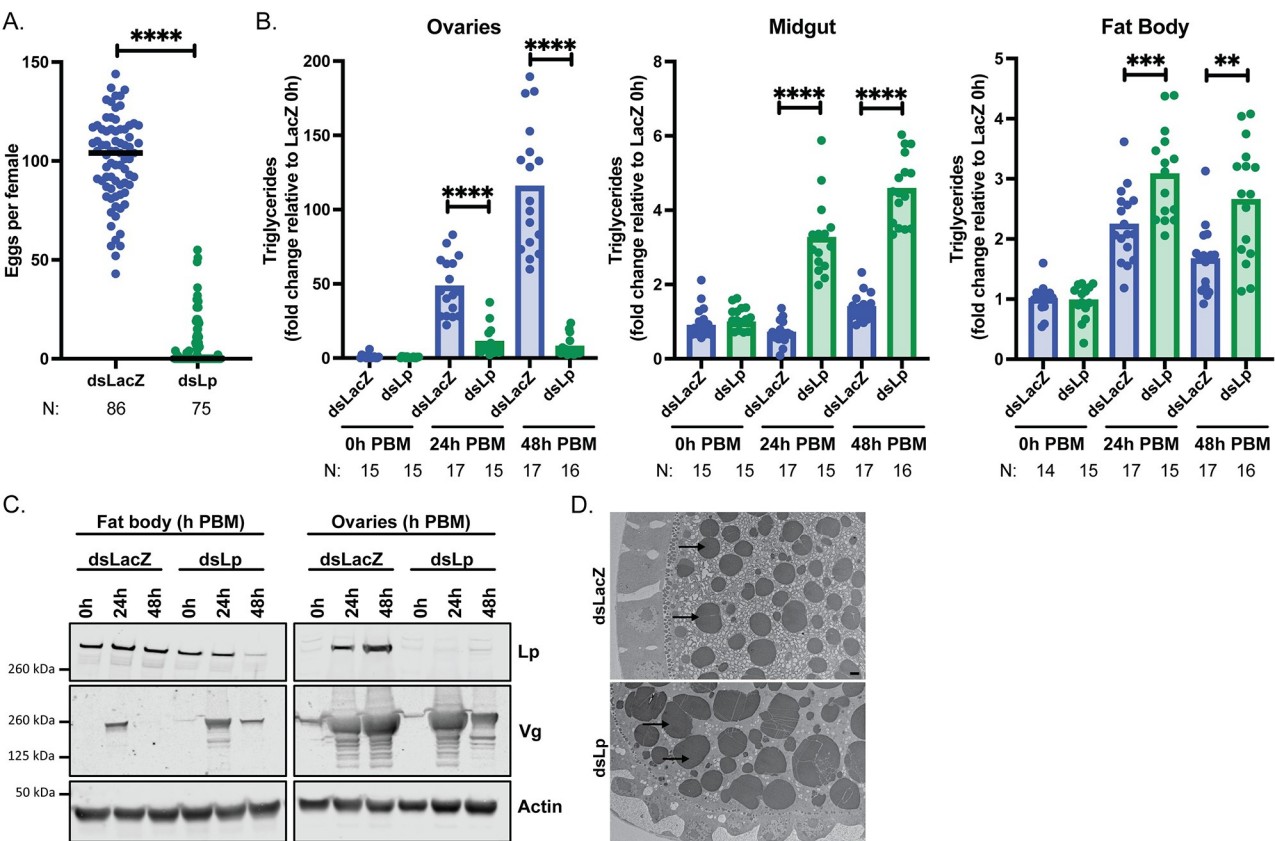

**Fig 1. *Lp* knockdown significantly impairs oogenesis and affects Vg localization.** (A) Following *Lp* knockdown, females develop a decreased number of eggs, as identified by dissection of ovaries of virgin females at 3–7 days PBM; each dot represents eggs per female; N = number of females, pooled from four biological replicates (Mann-Whitney: **** = p < 0.0001). (B) Triglycerides accumulate in the midgut and fat body of Lp-depleted females, and fail to accumulate in the ovaries; PBM = post blood meal; each dot represents tissues from three females; N = number of samples of three tissues, pooled from three biological replicates (REML variance component analysis by timepoint: **** = p < 0.0001; *** = p < 0.001). (C) Vg is persistently detectable in the fat body of Lp-depleted females and is decreased in the ovary at 48h PBM; samples of five tissues, representative blots (three biological replicates). (D) Transmission electron microscopy showing larger yolk granules (arrows pointing at darkly staining circles) in a ds*Lp* ovarian follicle compared to ds*LacZ* at 24h PBM; scale bar = 2 μm (one biological replicate).

control samples Vg was detected in the fat body at 24h PBM and was fully incorporated into the ovaries by 48h PBM, in Lp-depleted females Vg persisted in the fat body at the latter time point, perhaps due to impaired release, and its levels were reduced in the ovaries (Figs 1C and S1C). This was consistent with our observation that the ovaries of Lp-depleted females develop normally up to 24h PBM but degenerate by 48h PBM (S1D Fig). Moreover, very few mated *Lp*-knockdown females laid any eggs, and all were completely infertile (S1E Fig). Interestingly, as observed by electron microscopy, yolk granules in the few ovarian follicles that develop after *Lp* knockdown appeared larger at 24h PBM than those in control ovaries (Fig 1D), and Vg had decreased in intensity in Western blots by 48h PBM (Fig 1C), suggesting yolk degradation at this latter time point. We also determined levels of the steroid hormone 20E, given Lp has been shown to transport cholesterol [23], a key building block in ecdysteroid synthesis, and *Vg* expression is known to be induced by this ecdysteroid. However, when measured at 26h PBM (a peak time of 20E synthesis), we did not detect any difference in ecdysteroid levels in ds*Lp* females relative to controls (S1F Fig). A possible explanation for this surprising observation is that, after the first blood meal, ovary-synthesized E is produced from larval-derived cholesterol that is stored in ovaries and does not require Lp for transport.

Together, these data show that Lp-mediated shuttling of lipids from the midgut into the ovaries is an essential check point for oogenesis, as preventing this process results in misregulated vitellogenesis, most likely via 20E-independent pathways.

## *Vg* knockdown upregulates *Lp* expression and enhances lipid deposition into oocytes

We next assessed the role of *Vg* in egg development and fertility by knockdown of this gene in females prior to blood feeding (S2A Fig). As expected, given its main role during vitellogenesis, Vg depletion caused a significant reduction in number of eggs developed compared to control females (Fig 2A). More strikingly, however, knockdown of this yolk gene induced complete

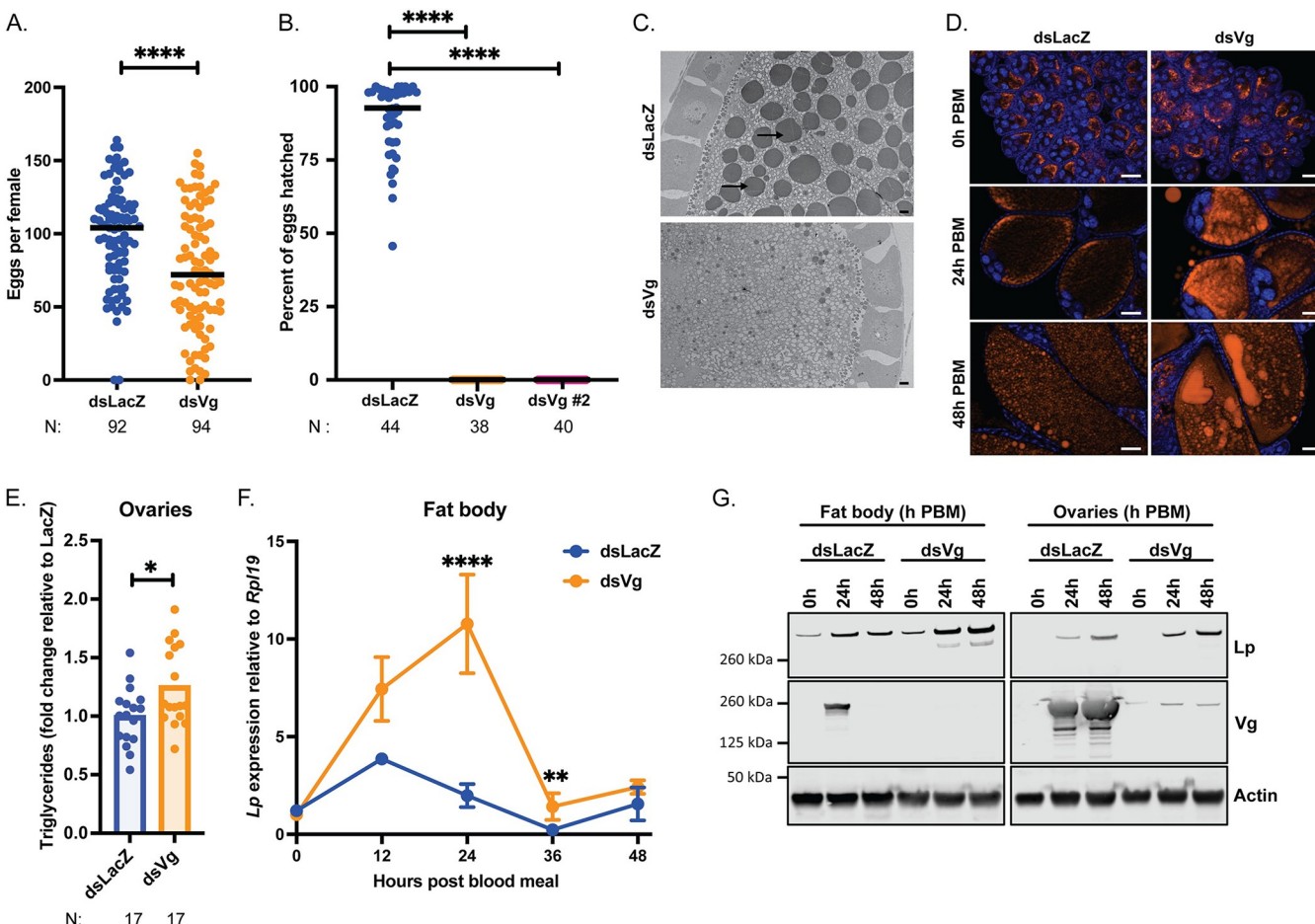

**Fig 2. *Vg* knockdown upregulates *Lp* expression and enhances lipid deposition into oocytes.** (A) Following *Vg* knockdown females develop fewer eggs, as identified by dissection of ovaries of virgin females at 3–7 days PBM; each dot represents eggs per female; N = number of females, pooled from five biological replicates (Mann-Whitney: **** = p < 0.0001). (B) *Vg* knockdown causes complete infertility of mated females; *Vg* was targeted by two different dsRNAs; each dot represents percent hatch rate per female; N = number of females, pooled from three biological replicates (Kruskal-Wallis with Dunn's multiple comparisons test: **** = p < 0.0001). (C) Transmission electron microscopy of developing oocytes showing a lack of yolk granules (arrows) upon *Vg* knockdown at 24h post blood meal; scale bar = 2 μm (representative images, two biological replicates). (D) Fluorescent microscopy showing an accumulation of neutral lipids (LD540, orange) in developing oocytes upon *Vg* knockdown; DNA (DAPI) in blue; scale bar = 50 μm (representative images, three biological replicates). (E) Triglyceride levels measured in ds*LacZ* and ds*Vg* ovaries at 48h post blood meal and normalized to mean ds*LacZ* levels in each replicate; each dot is representative of ovaries from three females; N = number of samples of three tissues, pooled from three biological replicates (Unpaired t-test: * = p < 0.05). (F-G) *Vg* knockdown results in an increase in Lp levels in the fat body at the mRNA (samples of ten tissues, four biological replicates (REML variance component analysis: ** = p < 0.01; **** = p < 0.0001)) (F) and protein (samples of five tissues, representative blots, three biological replicates) (G) levels in the fat body and ovaries.

infertility, with no larvae hatching from eggs oviposited by mated Vg-depleted mothers (Fig 2B). To confirm this phenotype, we tested a second dsRNA construct targeting *Vg*, and we again observed complete infertility (Fig 2B). No yolk bodies could be detected in ds*Vg* oocytes by electron microscopy (Fig 2C), and we also observed a notable decrease in total protein (S2B Fig) and free amino acids (S2C Fig) levels in ovaries, consistent with Vg being a major amino acid source.

Upon microscopic analysis, we noticed that developing Vg-depleted oocytes had a remarkable accumulation of neutral lipids at both time points analyzed (24 and 48h PBM) (Fig 2D), a finding which was supported by an assay showing higher triglyceride levels in ovaries (Fig 2E). Since transport of neutral lipids is mediated by Lp, we assayed *Lp* expression and found that this lipid transporter was upregulated at both transcript and protein levels (Fig 2F and 2G; S2D and S2E Fig). Specifically, while *Lp* transcripts in the fat body (the tissue where Lp is synthetized) peaked at 12h PBM in controls, after *Vg* knockdown they peaked later (at 24h PBM, the time of highest Vg expression under normal conditions) and at higher levels relative to controls (Figs 2F and S2D), paralleled by higher protein levels in the fat body and the ovaries (Figs 2G and S2E). Combined, these data suggest that Vg synthesis in the fat body and/or its incorporation into the ovaries is a signal that regulates *Lp* expression and prevents excessive Lp-mediated lipid accumulation into the developing oocytes, ensuring correct balance between nutrients and safeguarding fertility.

## *Vg* expression regulates Lp-mediated accumulation of lipids via TOR signaling

In *Ae. aegypti* mosquitoes, *Vg* transcription is partly regulated by TOR signalling [4,13]. As mentioned above, upon sensing amino acids derived from the blood meal, TOR phosphorylates S6K, which in turn activates AaGATAa, a transcription factor that binds the *Vg* promoter. We confirmed that *Vg* is regulated by TOR in *An. gambiae* by using the TOR inhibitor rapamycin, which decreased *Vg* expression in the fat body by almost 50% when applied to females at 2h PBM (Fig 3A). Surprisingly, we also noticed that *Vg* knockdown in turn affected TOR

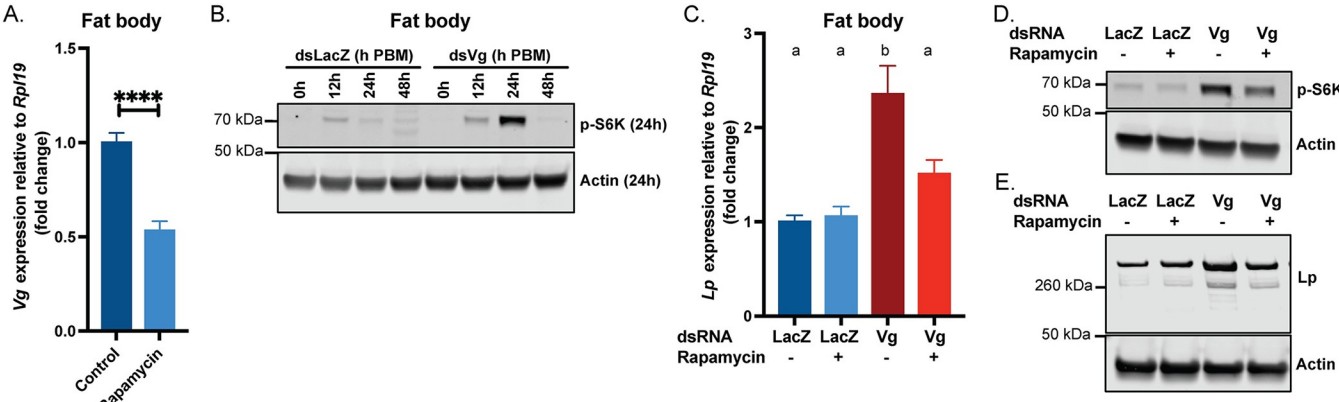

**Fig 3. Vg regulates *Lp* levels via TOR signaling.** (A) *Vg* expression relative to *Rpl19* is reduced in the fat body 24h PBM upon 0.5 μL of 40 μM rapamycin treatment 2h PBM; Control is 2.4% DMSO in acetone; all values normalized to control; samples of ten tissues, four biological replicates (unpaired t test: **** = p < 0.0001). (B) TOR signaling is induced in the fat body upon *Vg* knockdown as shown by Western blotting of phospho-S6K levels; samples of five tissues, representative blot (three biological replicates). (C) The *Vg*-knockdown mediated upregulation of *Lp* at 24h PBM is suppressed in the fat body of ds*Vg* females treated with 0.5 μL of 40 μM rapamycin 2h PBM; all values normalized to ds*LacZ* without rapamycin treatment; samples of ten tissues, four biological replicates (ANOVA). (D) Western blot showing phospho-S6K levels increase in the fat body at 24h PBM in Vg-depleted females, but this increase is repressed by rapamycin treatment; samples of five tissues, representative blot (four biological replicates). (E) Western blot showing fat body Lp levels are increased in *Vg*-knockdown females at 48h PBM, but this increase is repressed by rapamycin treatment; samples of five tissues, representative blot (four biological replicates).

signaling in the fat body, mediating its increase relative to controls (Figs 3B and S3A). Indeed, while in controls S6K phosphorylation had waned by 24h PBM, Vg-depleted females showed a strong signal at this time point, which corresponds to the peak of *Vg* expression. This was paralleled by significantly higher total protein levels after blood feeding (S3B Fig), indicating potential translational upregulation by TOR.

To determine whether TOR activation may be involved in the increased *Lp* expression observed in *Vg*-knockdown females, we treated Vg-depleted mosquitoes with rapamycin and assessed *Lp* levels relative to controls. Rapamycin treatment 2h PBM reduced S6K phosphorylation at 24h PBM following Vg depletion, confirming that this phosphorylation is indeed mediated by TOR (Figs 3D and S3C). *Lp* upregulation was also significantly reduced (at both protein and transcript levels) by rapamycin treatment, thereby implicating TOR signaling in the Vg-mediated regulation of this lipid transporter (Fig 3C and 3E; S3D Fig). In agreement with this observation, rapamycin treatment also slightly reduced the excess lipid accumulation observed upon *Vg* depletion (S3E Fig). Overall, these data suggest that in normal conditions *Vg* expression leads to repressed TOR signaling, putting a break on Lp synthesis thereby preventing excessive incorporation of lipids into the ovaries.

We hypothesized that excess amino acids not incorporated into Vg may be the triggers for activating TOR signaling. Indeed, in anopheline mosquitoes, yolk production utilizes up to 30% of the total protein content from the blood meal [24]. Consistent with this hypothesis, we detected a modest but significant increase in amino acid levels at 24h PBM in the fat body (but not in the hemolymph) of Vg-depleted females, which was amplified at 48h PBM (S3F and S3G Fig). TOR also incorporates inputs from ILPs to regulate metabolism [25], however expression levels of 7 *An. gambiae ILPs* were similar between groups at all time points analyzed (S3H Fig), likely ruling out insulin signaling as the cause of increased TOR signaling (note, ILP1 and 7, and ILP3 and 6 share sequence identity, resulting in 5 RT-qPCR plots).

Based on these data, a model emerges whereby in *An. gambiae* TOR signaling both controls *Vg* expression and is in turn regulated by amino acids not incorporated into this yolk protein, affecting the expression of *Lp* and the timely accumulation of lipids in the developing ovaries (summarized in the graphical abstract (S5 Fig)). Fine tuning of the function of these nutrient transporters is essential to support egg development and ensure fertility, demonstrating a previously unappreciated interplay that is key to the survival of this species.

### *Vg* knockdown causes early embryonic arrest

Our discovery that close cooperation between Lp and Vg is fundamental to fertility prompted us to dig deeper into the mechanisms causing lethality in embryos laid by Vg-depleted females. We found that most of these eggs were fertilized but not melanized (S4A Fig), and developing embryos did not reach the blastocyst stage, which control embryos instead reached at 3–5h post oviposition (Fig 4A). Some embryos were halted very early in development (Fig 4A, top panel), upon the first few mitotic divisions after the fusion of the gamete nuclei, while others reached the energid stage, where nuclei have divided and begin to migrate toward the outer edges (Fig 4A, bottom panel and zoomed-in insert). However, these nuclei displayed abnormal morphology, showing features reminiscent of apoptotic blebbing.

We did not detect any Vg in embryos from Vg-depleted mothers, contrary to controls and demonstrating that in early stages of embryogenesis Vg is entirely maternally derived (Figs 4B and S4B). Interestingly, we detected Lp in those embryos at very early stages (0–2h) post oviposition, suggestive of aberrant maternal deposition given Lp was not detected in controls immediately after oviposition but only appeared at later time points, after being expressed by the embryos (Figs 4B and S4B). Moreover, ds*Vg*-derived embryos showed significantly reduced

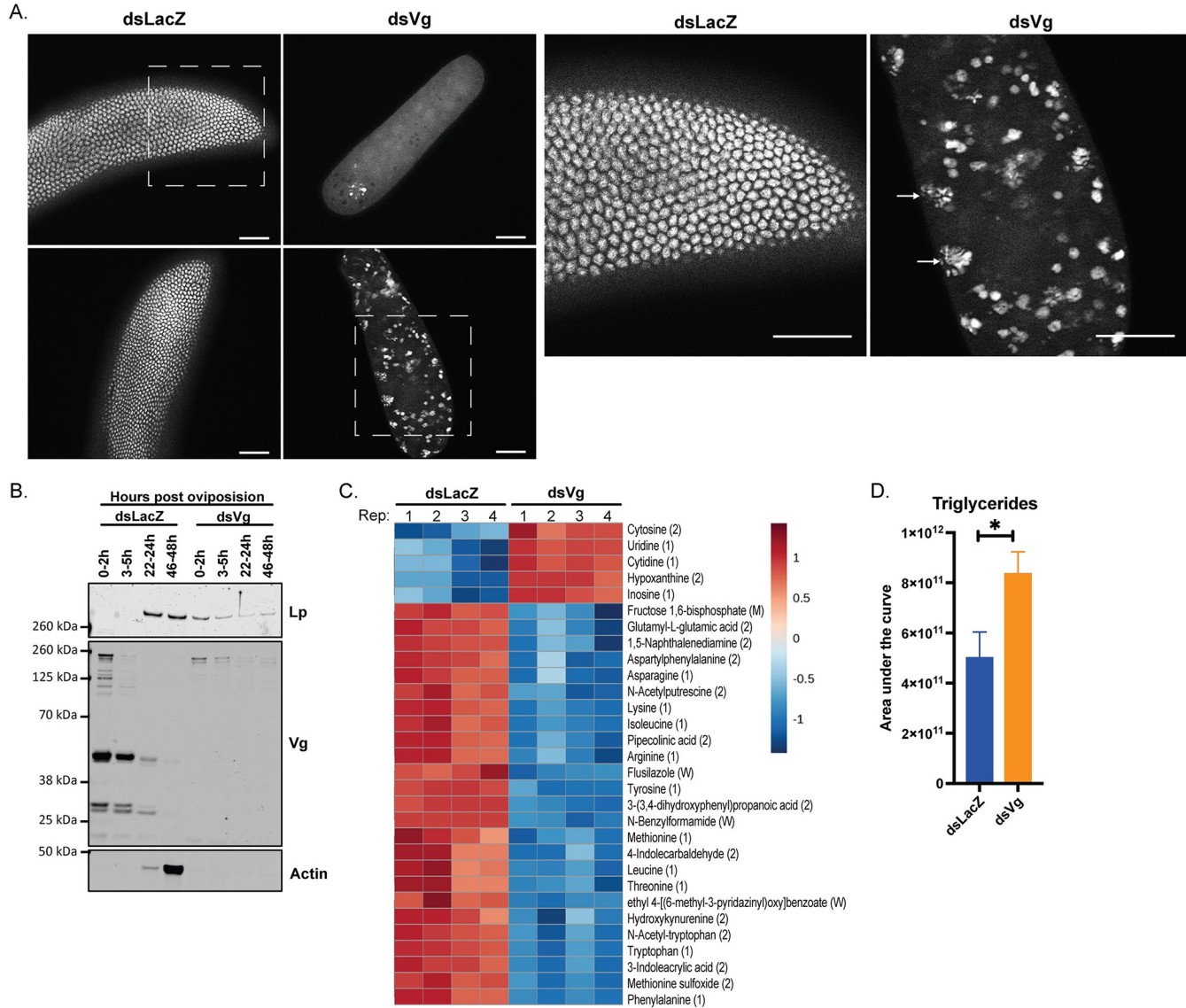

**Fig 4. *Vg* knockdown causes early embryonic arrest likely due to limited amino acids.** (A) DAPI staining of embryos from ds*LacZ* and ds*Vg* females at 3–5h post oviposition showing arrest of development before the cellular blastocyst stage in the ds*Vg*-derived embryos; scale bar = 50 μm (two biological replicates); dotted boxes represent the fields of view in the zoomed-in images to the right, and arrows point to nuclei characteristic of blebbing; representative images, two biological replicates; scale bar = 50 μm. (B) Lp is deposited into ds*Vg*-derived oocytes and not ds*LacZ*-derived ones, as observed by Western blot at specified times post oviposition; samples of 40 embryos, representative blot (three biological replicates). (C) Top 30 dysregulated metabolites (as determined by Metaboanalyst), of which 10 are amino acids, in 200 embryos derived from ds*Vg* mothers compared to those of ds*LacZ* 3–5h post oviposition (each column is representative of a biological replicate; four biological replicates). ID confidence is in brackets; from strongest to weakest: (1) = Level 1 ID, (2) = Level 2 ID, (M) = MasslistRT ID, (W) = Weak/Poor ID. (D) Triglyceride levels are elevated in ds*Vg*-derived embryos 3–5h post oviposition as determined by lipidomics; samples of 200 embryos, four biological replicates (Unpaired t test: * = p < 0.05).

protein synthesis confirming lack of development, as highlighted by undetectable actin expression over time (Fig 4B).

In agreement with these findings, metabolomic analysis of ds*Vg*-derived embryos (3–5h post oviposition) showed a depletion of amino acids, especially severe for those amino acids that are most prevalent in Vg (Fig 4C and S1 and S2 Tables). Intriguingly, the two amino acids more strikingly depleted, phenylalanine and tyrosine, are the starting compounds in the melanization pathway [26, 27], possibly explaining the lack of chorion darkening observed in ds*Vg*

embryos (S4A Fig). Also notable was the upregulation of free nucleotides as well as products of their catabolism such as hypoxanthine and inosine, likely due to impaired DNA replication leading to cell death (Fig 4C).

Given the observed incorporation of Lp into the embryo, we also performed lipidomic analysis at the same time point. Although there were no differences in total lipids in ds*Vg*-derived embryos relative to controls (S4C Fig and S3 Table), some minor lipid classes (phospholipids and glycerophospholipids) were decreased (S4D Fig), while we detected a remarkable increase in triglycerides, a subclass of neutral lipids, (Fig 4D), again consistent with the known role of Lp in transport of these lipids [10].

Therefore, when mothers are depleted of Vg, embryos cannot develop further than the first steps after fertilization, likely due to a severe drop in amino acid levels and protein synthesis.

## Discussion

Blood feeding is an essential process for the survival of mosquito species that, like *An. gambiae*, rely on blood nutrients for oogenesis. After a blood meal, different nutrient transporters start shuffling cholesterol, lipids and proteins from the midgut and fat body to the ovaries, and the individual roles of two of these factors, Lp and Vg, had been previously determined. There remained, however, important outstanding questions concerning whether and how these two essential transporters are co-regulated in order to ensure reproductive success. How do mosquitoes balance acquisition of different nutrients by the ovaries? What are the signals that limit lipid accumulation and that trigger vitellogenesis? Moreover, knowledge concerning Lp and Vg functions and their regulation in *Anopheles* as opposed to *Aedes* was only marginal. Here, we show that in these mosquitoes, oogenesis is the product of a precise and intricate interplay between these two factors (S5 Fig). The initial phase of oocyte development is dominated by Lp, which incorporates lipids (mostly triglycerides) and cholesterol into ovaries triggering their growth. During this early phase, as E is converted to 20E in the fat body, this steroid hormone triggers the expression of *Vg*, which peaks at 24h PBM and provides the oocytes with considerable stores of amino acids. When *Lp* expression is impaired, 20E levels are unchanged (S1F Fig) and yet egg development is strikingly reduced (Fig 1A), Vg localization and accumulation is affected (Fig 1C), and yolk bodies are aberrant (Fig 1D). Upon Lp depletion Vg seems to be retained in the fat body, since the ovaries are degenerating and cannot accumulate further yolk (S1D Fig). These findings suggest that correct lipid accumulation is an early check point that mosquitoes use to decide whether to proceed with vitellogenesis. Reducing *Vg* expression levels, on the other hand, leads to increased *Lp* expression (Fig 2F and 2G) and a surplus of glycerides in the ovaries (Fig 2E), which suggests that Vg synthesis is the signal that prevents excessive lipid trafficking by Lp into the oocytes. Interestingly, in another study *Vg* knockdown did not appear to affect *Lp* expression in *An. gambiae* females fed on mice infected with *P. berghei* parasites [20]. This discrepancy with our results may indicate that rodent malaria parasites, which are known to inflict severe reproductive costs in infected mosquito females [28], affect the normal accumulation of lipids during the oogenetic process, although other factors such as differences in temperature (infections with *P. berghei* are done at permissive temperatures around 20°C compared to standard mosquito rearing conditions of 28°C) cannot be excluded.

Strikingly, knockdown of *Vg* led to complete infertility. This phenotype was so penetrant that we confirmed it with a dsRNA second construct targeting a different region of the gene to rule out a possible unspecific effect of the first construct. This effect is reminiscent of observations in other insects, where altering Vg gene copy number, expression or internalization leads to complete sterility or decreased hatch rate by as yet unknown mechanisms [29–37].

Understanding how embryonic lethality is induced may lead to novel ideas for the design of mosquito-targeted interventions, so we set out to determine the mechanisms behind death. As *Vg* is also expressed in the female spermatheca after mating [38, 39] we initially thought that its depletion may have caused irreversible damage to sperm. The observation that eggs are fertilized and embryos start undergoing nuclear division (Fig 4A), however, appears to discount this possibility. Another potential explanation is that the improper deposition of lipid in the oocyte causes embryonic lethality, but based on our metabolomics analysis, a more plausible hypothesis is that lethality is a result of amino acid starvation. We show that embryos from Vg-depleted females are significantly deficient in 14 of the 19 identified amino acids, which results in a lack of building blocks for translation and thus development (S1 Table). It is plausible to speculate that depletion in these essential nutrients may activate the amino acid starvation response pathway, triggering a global shutdown of translation that may lead to apoptosis, compatible with our observation of nuclei blebbing in those embryos [40–42].

Our data using rapamycin suggest that the *An. gambiae* Vg is regulated by TOR (Fig 3A), as was previously shown in *Ae. aegypti* [15]. Surprisingly, however, our findings also suggest that Vg expression leads to suppressed TOR signaling, as upon *Vg* knockdown S6K phosphorylation was strongly upregulated possibly due to an increase in free amino acids (Figs 3B and S3F). This upregulation in TOR signaling also resulted in an increase in Lp transcription and translation, further shedding light on regulation of *Lp* expression. Previous studies had shown that *Lp* levels in *An. gambiae* are under steroid hormone control, as impairing 20E signaling caused an increase in *Lp* transcription at 24h, 36h and 48h PBM [22]. Since Vg is under 20E control, it is probable that the result observed by Werling *et al.* at 24h PBM (increased *Lp* expression in ds*EcR*) is mediated by reduced Vg expression/increased TOR signaling, while additional EcR-controlled mediators, or EcR itself, are responsible for the decreased *Lp* expression at later timepoints. With the caveat that our current results were obtained only by using the inhibitor rapamycin rather than by also depleting key components of the TOR pathway, these combined observations may suggest that TOR and 20E signaling exert opposite effects on *Lp* expression—with 20E repressing its levels and TOR enhancing them, an intriguing finding that deserves more thorough investigation in future studies. Compatible with our data, the Lp promoter has putative GATA transcription factor binding motifs, some of which are known to be regulated by TOR signaling [17,43].

Does the interplay between Lp and Vg also affect the development of *P. falciparum* parasites? An earlier study showed that following *Lp* knockdown in *An. gambiae*, *P. falciparum* oocyst numbers are decreased [22]. No other effects were detected on parasite development, unlike in the mouse malaria parasite *P. berghei* where Lp depletion, besides a decrease in oocyst numbers, also led to reduced oocyst growth [20]. While the role of Vg in *P. falciparum* has not been directly determined, it is known that impairing 20E signaling (which in turn negatively affects *Vg* levels) has profound and opposite effects on parasites, as it reduces parasite numbers but accelerates their growth. Regardless of its impact on parasite development, our data reveal the interplay between Lp and Vg as essential for mosquito fertility, opening the possibility of targeting it to reduce the reproductive success of mosquito populations.

## Materials and methods

### Mosquito lines and rearing

G3 *Anopheles gambiae* mosquitoes were reared at 27˚C, 70–80% humidity. Adults were fed 10% glucose solution and purchased human blood (Research Blood Components, Boston, MA). Females and males were separated by pupae sexing, and females were kept separate to

ensure virgin status or mixed with males at a 1:2 ratio for fertility experiments and egg collections.

### dsRNA generation

A 816bp *LacZ* fragment and 600bp *Lp* (AGAP001826) fragment were generated from plasmids pLL100-LacZ and pLL10-Lp as described previously [22, 44, 45] using T7 primer (5'–TAA-TACGACTCACTATAGGG–3'). A 552 bp fragment of *Vg* (AGAP004203) corresponding to bases 3374–3925 of the *Vg* cDNA was amplified from plasmid pLL10-Vg, a gift from Miranda Whitten and Elena A. Levashina (Max Planck Institute for Infection Biology, Berlin), using a primer matching the inverted T7 promoters (same as above). To generate the ds*Vg* #2 construct, a 284bp PCR product was generated from *An. gambiae Vg* cDNA (AGAP004203) corresponding to bases 4530–4813 using forward primer 5'–ATTGGGTACCGGGCCCCCCCG CACGTCTCGATGAAGGGTA–3' and reverse primer 5'–GGGCCGCGGTGGCGGCCGC TCTAGACCTGCCCTGGAAGAAGTAGTCC–3'. The pLL10-Vg backbone and the PCR fragment were restriction digested with XbaI and XhoI, separated on an agarose gel and gel purified. Then, fragments were assembled using NEBuilder HiFi DNA Assembly Kit. PCR product was amplified using T7 primer. A 495 bp eGFP fragment was amplified from plasmid pCR2.1-eGFP using pCR2.1-T7F: 5'–taatacgactcactatagggCCGCCAGTGTGCTGGAA–3' and pCR2.1-T7R: 5'–taatacgactcactatagggGGATATCTGCAGAATTCGCCC–3' as described previously [46]. PCR for dsRNA generation was separated by gel electrophoresis for size confirmation, and transcribed into dsRNA by *in vitro* transcription Megascript T7 kit (ThermoFisher Scientific) [22]. dsRNA was purified by phenol-chloroform extraction, and diluted to 10 μg/μL.

### dsRNA injections

Females on day 1 post eclosion were injected with 69 nL of dsRNA (ds*LacZ*, ds*Vg*, ds*Vg* #2, ds*Lp* ds*GFP*) using Nanoject III (Drummond), and allowed to recover. Surviving females were fed with blood 3 days post injection. Unfed females were removed from experimental cages.

### Egg counts

Virgin females were dissected 3–7 days PBM, and the egg clutches were counted. Although eggs take 2–3 days to fully develop after blood feeding, virgin females do not oviposit their eggs, and once developed, the number of eggs does not change from day 3 to 7 PBM in our laboratory conditions, making it possible to count egg numbers even at 7 days PBM.

### Fertility assay

Injected females were mixed with males at a 1:2 ratio immediately after injection, and blood fed three days later. One day after blood feeding, fed females were moved to individual cups with around 2cm of water at the bottom. Hatched and unhatched eggs from every cup were counted within a week.

### RNA extraction, cDNA synthesis and RT-qPCR

Fat bodies or heads (10 tissues per tube) from female mosquitoes were dissected in PBS and stored at -80˚C in 300 μL TRI Reagent (ThermoFisher Scientific). Samples were thawed and bead beaten using 2 mm beads. Then RNA was extracted using manufacturer's instructions with a modification to wash the RNA pellet using 70% ethanol. 2.5 μg of RNA was aliquoted and DNase treated with Turbo DNase from the TURBO DNA-free Kit (ThermoFisher

Scientific), followed by DNase inactivation from the same kit. cDNA synthesis was carried out in 100 μL reactions using random primers (ThermoFisher Scientific), dNTPs (ThermoFisher Scientific), first strand buffer (VWR), RNAseOUT (ThermoFisher Scientific) and MMLV (ThermoFisher Scientific) [22]. Relative quantification RT-qPCR was carried out using SYBR--Green mix and primers from

S4 Table. Primers were designed on exon-exon junctions where possible. Quantification was performed in triplicate using the QuantStudio 6 Pro qPCR machine (ThermoFisher Scientific). *Rpl19* was used as the endogenous control for relative quantification.

## Ecdysteroid level measurement

Ten females at 26h PBM were collected per sample by removing their heads and storing them in 400 μL of 100% methanol at -80˚C. Ecdysteroids were measured using the 20E enzyme immunoassay kit (Cayman Chemical), according to manufacturer instructions and as described in previously published work [22]. Although the kit is targeted at identifying 20E, the immunoassay cross-reacts with other ecdysteroids. Hence, the measurements in S2 Fig are labelled as "Ecdysteroid levels".

## Immunofluorescence microscopy and tissue staining

Ovaries were dissected from females at specified timepoints and incubated at room temperature in 4% paraformaldehyde (PFA) for 30 minutes, followed by 3 washes in PBS for 15 minutes. Ovaries were permeabilized and blocked with 0.1% Triton X-100 and 1% bovine serum albumin (BSA) in PBS for 1h followed by 3 washes in PBS for 15 minutes. Ovaries were then incubated with DAPI and LD540 [47], both at a concentration of 1 μg/mL, at room temperature for 15 minutes. After staining, ovaries were washed in PBS 3 times for 15 minutes and mounted using Vectashield mounting medium (Vector Laboratories). Images were captured on a Zeiss Inverted Observer Z1.

## Embryo collections for microscopy

An egg bowl was inserted into cages of mated females blood fed 96h before. The egg bowl was removed 2h later, and embryos were collected 3h later, resulting in a timepoint of 3–5h. Embryos were dechorionated and cracked as described previously [48]. Briefly, embryos were washed with 25% household bleach (2% sodium hypochlorite final concentration) in 1xPBS, collected into glass vials with 9% PFA and heptane, and rotated for 25 minutes. PFA was removed and replaced with deionized water twice. Vials were shaken for another 30 minutes. Then water was replaced with boiling water and incubated in a hot water bath for 30 seconds, and immediately replaced with ice cold water. Both water and heptane were removed and replaced with heptane and methanol. Embryos were swirled vigorously to crack the shell, washed 3 times with methanol and collected into methanol. Embryos were then coaxed out of eggshells [49], and stained with DAPI as described above.

## Transmission electron microscopy

Dissected mosquito ovaries were collected in 200 uL of fixative (2.5% paraformaldehyde, 5% glutaraldehyde, 0.06% picric acid in 0.2M cacodylate buffer) and spun down briefly to fully submerge the tissues in fixative. Fixed samples were submitted to the Harvard Medical School Electron Microscopy Core. Samples were washed once in 0.1M cacodylate buffer, twice in water, and then postfixed with 1% osmium tetroxide/1.5% potassium ferrocyanide in water for 1h. Samples were then washed twice in water followed by once in 50 mM maleate buffer pH

5.15 (MB). Next, the samples were incubated for 1h in 1% uranyl acetate in MB, followed by one wash in MB, and two washes in water. Samples were then subjected to dehydration via an increasing ethanol gradient (50%, 70%, 90%, 100%, 100% ethanol) for 10 minutes each. After dehydration, samples were placed in propylene oxide for 1h and then infiltrated overnight in a 1:1 mixture of propylene oxide and TAAB 812 Resin (https://taab.co.uk/, #T022). The following day, samples were embedded in TAAB 812 Resin and polymerized at 60˚C for 48h. Ultra-thin sections (roughly 80 nm) were cut on a Reichert Ultracut-S microtome, sections were picked up onto copper grids, stained with lead citrate and imaged in a JEOL 1200EX transmission electron microscope equipped with an AMT 2K CCD camera.

### Rapamycin treatment

Rapamycin was dissolved in DMSO at 10 mM. DMSO-Rapamycin solution was then mixed with acetone as a volatile carrier with the final concentrations being 40 μM rapamycin and 2.4% DMSO (v/v). 0.5 μL was topically applied to the posterior thoraces of females on ice at 2h PBM. Control mosquitoes were treated with 0.5 μL of 2.4% DMSO in acetone. Acetone was used as a volatile carrier to ensure even delivery of the compound through the cuticle. Mosquitoes were placed into cages with 10% glucose to recover.

### Hemolymph collections

For amino acid assay, hemolymph was collected by making a tear between the last and second-to-last segments on the abdomen, and injecting 2 μL of purified deionized water into each mosquito. A drop of liquid was collected from the abdomen with a pipette. Hemolymph from five females was pooled for the amino acid assay.

### Primary antibodies

Antibody against Vg was generated with Genscript by injecting a rabbit with a Vg peptide (QADYDQDFQTADVKC). Rabbit serum was affinity purified to produce a polyclonal antibody used at 1:1000 for Western blotting.

Anti-Lp antibody was also generated with Genscript using the following peptide: FQRDASTKDEKRSGC [22]. This antibody was used at 1:4000 for Western blotting.

Anti-actin antibody was acquired from Abcam (MAC237) and used at a dilution of 1:4000.

Phospho-S6K antibody was acquired from MilliporeSigma (07-018-I) and used at a dilution of 1:1000.

### Western blotting

5 tissues per sample, or 40 embryos were collected into 55 μL of PBS with protease and phosphatase inhibitors (cOmplete Mini EDTA free protease inhibitor cocktail, Halt phosphatase inhibitor). DTT (200 mM) and NuPAGE LDS Sample Buffer were added. Tissues were bead beaten and boiled for 10 minutes. Then, 1/10th of the sample was loaded onto either a NuPAGE 4–12% Bis-Tris gel or NuPAGE 3–8% Tris Acetate gel (when blotting for Lp). Gels were transferred for 10 minutes at 22V using an iBlot2 machine and iBlot2 PVDF Stacks, blocked in Intercept Blocking Buffer (LI-COR) for 1h, and then incubated with antibody overnight at 4˚C. Membranes were washed with PBS-T 4 times for 5 minutes, and incubated with LI-COR secondary antibodies (Goat anti-Rat 680LT; Donkey anti-Rabbit 800CW) for 1–2h. Membranes were washed again with PBS-T 4 times and once in PBS. Membranes were imaged using a LI-COR developer. Western blot bands were quantified using ImageJ, with pixel intensities being normalized to the loading control (actin) as well as background. During embryonic

development, actin levels rapidly change, so for embryo samples, Lp and Vg were normalized only to background.

## Triglyceride assay and Bradford assay

Three tissues per sample were collected into NP40 Assay Reagent, and Triglyceride Colorimetric Assay was performed according to manufacturer's instructions (Cayman Chemical). Briefly, tissues were homogenized by bead beating in 32 μL of the NP40 Assay Reagent, centrifuged at 10,000$g$ for 10 minutes. 10 μL of supernatant was added to 150 μL of Enzyme Mixture in duplicate and incubated for 30 minutes at 37˚C. Absorbance was measured at 530 nm.

Of note, the kit releases glycerol from triglycerides and measures glycerol levels, and does not measure triglyceride levels directly.

The same supernatant was also used for Bradford assay. Supernatant was diluted 1 in 10 and 4 μL of diluted supernatant was added to 200 μL of Bradford reagent (Bio-Rad) at room temperature, and absorbance was recorded at 595 nm.

## Amino acid assay

Five tissues per sample were collected into 50 μL of Ultrapure Distilled Water (Invitrogen), and amino acid assay was performed according to manufacturer's instructions (EnzyChrom L-Amino Acid Assay Kit ELAA-100). Briefly, samples were homogenized by bead beating and centrifuged at 10,000$g$ for 15 minutes. 20 μL of supernatant was mixed with Working Reagent in duplicate and incubated at room temperature for 60 minutes. Absorbance was recorded at 570 nm.

## Metabolomics and lipidomics

### Sample collection

200 embryos were collected into 1 mL of methanol and homogenized by bead beating with five 2 mm glass beads, then transferred to 8 mL glass vials. Tubes were then rinsed with 1 mL of methanol that was pooled with the homogenized sample, and 4 mL of cold chloroform was added to the glass vials, which were then vortexed for 1 minute. 2 mL of water was added and glass vials were vortexed for another minute. Vials were then centrifuged for 10 minutes at 3000 $g$. The upper aqueous phase was submitted for metabolomics, and lower chloroform phase was submitted for lipidomics to the Harvard Center for Mass Spectrometry.

### Metabolomics mass spectrometry

Samples were dried down under Nitrogen flow and resuspended in 25 μL of acetonitrile 30% in water. Ten microliter of each sample was used to create a pool sample for MS2/MS3 data acquisition. Samples were analyzed by LC-MS on a Vanquish LC coupled to an ID-X MS (Thermofisher Scientific). Five μL of sample was injected on a ZIC-pHILIC peek-coated column (150 mm x 2.1 mm, 5 μm particles, maintained at 40˚C, SigmaAldrich). Buffer A was 20 mM Ammonium Carbonate, 0.1% Ammonium hydroxide in water and Buffer B was Acetonitrile 97% in water. The LC program was as follow: starting at 93% B, to 40% B in 19 min, then to 0% B in 9 min, maintained at 0% B for 5 min, then back to 93% B in 3 min and re-equilibrated at 93% B for 9 min. The flow rate was maintained at 0.15 mL min$^{-1}$, except for the first 30 seconds where the flow rate was uniformly ramped from 0.05 to 0.15 mL min$^{-1}$. Data was acquired on the ID-X in switching polarities at 120000 resolution, with an AGC target of 1e5, and a m/z range of 65 to 1000. MS1 data was acquired in switching polarities for all samples. MS2 and MS3 data was acquired on the pool sample using the AquirX DeepScan function,

with 5 reinjections, separately in positive and negative ion mode. Data was analyzed in Compound Discoverer software (CD,version 3.3 Thermofisher Scientific). Identification was based on MS2/MS3 matching with a local MS2/MS3 mzvault library and corresponding retention time built with pure standards (Level 1 identification), or on mzcloud match (level 2 identification). Compounds where the retention time and the accurate mass matched an available standard, but for which MS2 data was not acquired are labelled MasslistRT matches. Each match was manually inspected.

Metabolomics heatmap was generated using Metaboanalyst 5.0 by log 10 transforming the area under the curve values for metabolites identified as described above.

## Lipidomics mass spectrometry

Samples were dried down under Nitrogen flow and resuspended in 60 μL of chloroform. LC–MS analyses were modified from [50] and were performed on an Orbitrap QExactive plus (Thermo Scientific) in line with an Ultimate 3000 LC (Thermo Scientific). Each sample was analyzed separately in positive and negative modes, in top 5 automatic data dependent MSMS mode. Twenty μL of sample was injected on a Biobond C4 column ($4.6 \times 50$ mm, 5 μm, Dikma Technologies, coupled with a C4 guard column). Flow rate was set to 100 μl min$^{-1}$ for 5 min with 0% mobile phase B (MB), then switched to 400 μl min$^{-1}$ for 50 min, with a linear gradient of MB from 20% to 100%. The column was then washed at 500 μl min$^{-1}$ for 8 min at 100% MB before being re-equilibrated for 7min at 0% MB and 500 μl min$^{-1}$. For positive mode runs, buffers consisted for mobile phase A (MA) of 5 mM ammonium formate, 0.1% formic acid and 5% methanol in water, and for MB of 5 mM ammonium formate, 0.1% formic acid, 5% water, 35% methanol in isopropanol. For negative runs, buffers consisted for MA of 0.03% ammonium hydroxide, 5% methanol in water, and for MB of 0.03% ammonium hydroxide, 5% water, 35% methanol in isopropanol. Lipids were identified and integrated using the Lipidsearch software (version 4.2.27, Mitsui Knowledge Industry, University of Tokyo). Integrations and peak quality were curated manually before exporting and analyzing the data in Microsoft Excel.

## Quantification and statistical analyses

All statistical tests were performed in GraphPad Prism 9.0 and JMP 17 Pro statistical software. The number of replicates and statistical tests performed are mentioned in the figure legend. Detailed outputs of statistical models and numerical data are provided in the supporting information (S5–S7 Tables). Residual Maximal Likelihood (REML) variance components analysis was used by fitting linear mixed models after data transformation to resemble normality. dsRNA treatment and timepoint were included as fixed effects and replicate as a random effect. If transformation was not possible, a generalized linear model was used instead. Interaction terms were removed when not significant and models with lower AICc scores were kept. Multiple comparisons were calculated using pairwise Student's t-tests at each timepoint followed by FDR correction at a 0.05 significance level.

## Supporting information

**S1 Fig. *Lp* knockdown significantly impairs oogenesis and affects Vg expression and localization.** (A) Successful *Lp* knockdown as determined by RT-qPCR of *Lp* expression levels relative to *Rpl19* in the fat body of ds*LacZ* and ds*Lp* females; samples of ten tissues, three biological replicates (REML variance component analysis: * = p < 0.05; **** = p < 0.0001; three biological replicates). (B) RT-qPCR of *Vg* expression levels relative to *Rpl19* in the fat body of ds*LacZ* and ds*Lp* females; samples of ten tissues, four biological replicates (REML variance component

analysis: ** = p < 0.01; **** = p < 0.0001; four biological replicates). (C) Western blot quantification from Fig 1C showing an accumulation of Vg in the fat body and a decrease of Vg in the ovaries upon *Lp* knockdown; samples of five tissues, three biological replicates (REML variance component analysis: ** = p < 0.01; **** = p < 0.0001). (D) Images of ovaries at 24 and 48h post blood meal showing that Lp depleted ovaries develop normally at first before degenerating by 48h; representative images, three biological replicates; scale bar = 2 mm. (E) *Lp* knockdown causes complete infertility of mated females that lay eggs; each dot represents percent hatch rate per female; N = number of females, pooled from two biological replicates (Mann-Whitney: **** = p < 0.0001). Very few (3 of 71) mated blood fed *Lp*-knockdown females laid eggs, unlike controls (60 of 64). (F) There is no difference in ecdysteroid levels of whole female bodies at 26h PBM upon *Lp* knockdown; each dot represents ecdysteroid level per female derived from a sample of 10 females (Unpaired t-test: not significant; three biological replicates of three samples each).
(TIF)

**S2 Fig. *Vg* knockdown effects in fat body and ovaries.** (A) Successful *Vg* knockdown as determined by RT-qPCR of *Vg* expression levels relative to *Rpl19* in the fat body; samples of ten tissues, three biological replicates (REML variance component analysis: * = p < 0.05; **** = p < 0.0001). (B) Fold change in protein levels measured by Bradford assay in the ovaries of ds*LacZ* and ds*Vg* females before blood meal and at 24h and 48h post blood meal (PBM); each dot is representative of three pairs of ovaries; N = number of samples of three tissues, pooled from three biological replicates (REML variance component analysis by timepoint: ** = p < 0.01; **** = p < 0.0001). (C) Fold change in free amino acid levels in the ovaries of ds*LacZ* and ds*Vg* females before blood meal and at 24h and 48h PBM; each dot is representative of five pairs of ovaries; N = number of samples of five tissues, pooled from three biological replicates (REML variance component analysis by timepoint: * = p < 0.05). (D) *Vg* knockdown by a second *Vg* RNAi fragment also results in an increase in *Lp* levels as determined by RT-qPCR; samples of ten tissues, three biological replicates (REML variance component analysis: * = p < 0.05). (E) Western blot quantification from Fig 2G showing an accumulation of Lp in the fat body and ovaries upon *Vg* knockdown; samples of five tissues, three biological replicates (REML variance component analysis: fat body–dsRNA: p < 0.01; * = p < 0.05; ovaries–dsRNA: p < 0.05; * = p < 0.05).
(TIF)

**S3 Fig. *Vg* expression regulates Lp-mediated accumulation of lipids via TOR signaling.** (A) Western blot quantification from Fig 3B showing an increase in phospho-S6K levels in the fat body upon *Vg* knockdown; samples of five tissues, three biological replicates (REML variance component analysis: **** = p < 0.0001). (B) Fold change in protein levels, measured by Bradford assay of the fat body are increased in ds*Vg* females; each dot is representative of three fat bodies; N = number of samples of three tissues, pooled from three biological replicates (REML variance component analysis by timepoint: **** = p < 0.0001). (C) Western blot quantification from Fig 3D showing a decrease in phospho-S6K levels in the fat body upon rapamycin treatment; samples of five tissues, three biological replicates (ANOVA). (D) Western blot quantification from Fig 3E showing Lp protein levels upon *Vg* knockdown and rapamycin treatment; samples of five tissues, four biological replicates (ANOVA). (E) Triglyceride levels measured in ds*LacZ* and ds*Vg* ovaries upon 0.5 μL of 40 μM rapamycin treatment at 72h PBM and normalized to mean ds*LacZ* levels in each replicate; each dot is representative of ovaries pooled from three females; N = number of samples of three tissues, pooled from two biological replicates (ANOVA). (F) Fold change in free amino acid levels in the fat bodies of ds*LacZ* and ds*Vg* females before blood meal and at 12h, 24h and 48h PBM; each dot is representative of five

ovaries; N = number of samples of five tissues, pooled from three biological replicates (REML variance component analysis by timepoint: ** = p < 0.01). (G) Fold change in free amino acid levels in the hemolymph of ds*LacZ* and ds*Vg* females before blood meal and at 12h, 24h and 48h PBM; each dot is representative of hemolymph collected from five females; N = number of samples of five hemolymphs, pooled from three biological replicates (REML variance component analysis by timepoint). (H) RT-qPCR of *ILP* expression levels relative to *Rpl19* in the heads of ds*LacZ* and ds*Vg* females; samples of ten tissues, ILP1/7: two biological replicates; other ILPs: three biological replicates (REML variance component analysis).
(TIF)

**S4 Fig. *Vg* knockdown in females prevents embryo melanization and causes early embryonic arrest.** (A) Light microscopy of embryos from ds*LacZ*- and ds*Vg*-derived females at 3–5h post oviposition; scale bar = 200 μm. (B) Western blot quantification of Lp and total Vg from Fig 4B; samples of 40 embryos, three biological replicates (REML variance component analysis: **** = p < 0.0001; ** = p < 0.01; * = p < 0.05). (C-D) Total lipids (C) and lipid classes (D) in ds*LacZ*- and ds*Vg*-derived 200 embryos 3–5h post oviposition as determined by lipidomics, four biological replicates (Unpaired t tests, followed by FDR correction: * = p < 0.05; ** = p < 0.01).
(TIF)

**S5 Fig. Graphical abstract.**
(TIF)

**S1 Table. Vg amino acids and their decrease in embryos upon *Vg* depletion.**
(DOCX)

**S2 Table. Metabolomics dataset of ds*LacZ* and ds*Vg* embryos 3–5h post oviposition.**
(XLSX)

**S3 Table. Lipidomics dataset of ds*LacZ* and ds*Vg* embryos 3–5h post oviposition.**
(XLSX)

**S4 Table. RT-qPCR primer sequences.**
(DOCX)

**S5 Table. Details of statistical tests and outputs.** For RT-qPCR, at least three independent biological replicates of a gene expression timecourse were analyzed, except for ILP1, where one replicate was excluded as an outlier. Effect test outputs are reported here. Multiple comparisons were calculated using pairwise Student's t tests at each timepoint followed by FDR correction (see S6 Table). KD = knock down; rand = random effect; FDR = false discovery rate.
(DOCX)

**S6 Table. Details of post-hoc statistical testing.** Tests were conducted for significant differences using an FDR of 0.05. See S5 Table.
(DOCX)

**S7 Table. Numerical data plotted in graphs.**
(XLSX)

## Acknowledgments

We thank Kate Thornburg, Emily Selland, Elizabeth Nelson, Kaileigh Bumpus, and Aaron Stanton for rearing mosquitoes used in this study. We thank all other members of the

Catteruccia lab for their ideas and feedback, as well as Krystle Kalafut for her insights on TOR signalling. We thank Christoph Thiele for providing the LD540 stain. Electron Microscopy Imaging, consultation and services were performed in the HMS Electron Microscopy Facility, and we thank Maria Ericsson, Peg Coughlin, and Anja Nordstrom for their help.

## Author Contributions

**Conceptualization:** Iryna Stryapunina, Maurice A. Itoe, W. Robert Shaw, Flaminia Catteruccia.

**Data curation:** Iryna Stryapunina, Maurice A. Itoe, Charles Vidoudez.

**Formal analysis:** Iryna Stryapunina, Maurice A. Itoe, Charles Vidoudez, W. Robert Shaw, Flaminia Catteruccia.

**Funding acquisition:** Iryna Stryapunina, Maurice A. Itoe, Flaminia Catteruccia.

**Investigation:** Iryna Stryapunina, Maurice A. Itoe, Queenie Trinh, Charles Vidoudez, Esrah Du, Lydia Mendoza, Oleksandr Hulai, Jamie Kauffman, John Carew, W. Robert Shaw, Flaminia Catteruccia.

**Methodology:** Iryna Stryapunina, Maurice A. Itoe, Charles Vidoudez, W. Robert Shaw, Flaminia Catteruccia.

**Project administration:** Iryna Stryapunina, Maurice A. Itoe, W. Robert Shaw, Flaminia Catteruccia.

**Resources:** Charles Vidoudez, Flaminia Catteruccia.

**Software:** Charles Vidoudez, W. Robert Shaw.

**Supervision:** W. Robert Shaw, Flaminia Catteruccia.

**Validation:** Iryna Stryapunina, Charles Vidoudez, W. Robert Shaw, Flaminia Catteruccia.

**Visualization:** Iryna Stryapunina, W. Robert Shaw, Flaminia Catteruccia.

**Writing – original draft:** Iryna Stryapunina, Maurice A. Itoe, Charles Vidoudez, Esrah Du, W. Robert Shaw, Flaminia Catteruccia.

**Writing – review & editing:** Iryna Stryapunina, W. Robert Shaw, Flaminia Catteruccia.

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
