## [Decision Letter · Decision Letter 0]

8 Sep 2023

Dear Dr Catteruccia,

Thank you very much for submitting your Research Article entitled 'Interplay between nutrient transporters ensures fertility in the malaria mosquito *Anopheles gambiae*' to PLOS Genetics.

The manuscript was fully evaluated at the editorial level and by independent peer reviewers. The reviewers appreciated the attention to an important problem, but raised some substantial concerns about the current manuscript. Based on the reviews, we will not be able to accept this version of the manuscript, but we would be willing to review a much-revised version. We cannot, of course, promise publication at that time.

If you decide to revise the manuscript for further consideration at PLOS Genetics, please aim to resubmit within the next 60 days, unless it will take extra time to address the concerns of the reviewers, in which case we would appreciate an expected resubmission date by email to plosgenetics@plos.org.

We are sorry that we cannot be more positive about your manuscript at this stage. Please do not hesitate to contact us if you have any concerns or questions.

Yours sincerely,

Takaaki Daimon

Academic Editor

PLOS Genetics

Gregory P. Copenhaver

Editor-in-Chief

PLOS Genetics

Reviewer's Responses to Questions

**Comments to the Authors:**

Reviewer #1: The manuscript "Interplay between nutrient transporters ensures fertility in the malaria mosquito Anopheles gambiae" presented by Stryapunina and collaborators, examines the importance of Vg and Lp to ensure the reproduction of A. gambiae. The work is well done, with solid data, but some experiments or results have confusing descriptions of oocytes, egg chambers, eggs, or embryos, making it difficult to understand certain parts. Issues that can be improved by revising the manuscript. The authors propose to demonstrate the possible control that Vg and Lp exert on each other to control the reproduction of this mosquito species. To do this, molecular, microscopic, and analytical tools were used. Demonstrating at the end that the two proteins are necessary to complete the reproduction of A. gambie, as it happens with other insect species, but did not finish defining how the interaction between these two proteins occurs. With the work done using different methodologies, the authors can have more data and detail about what happens in the ovaries or fat body of the treated females, information that would improve the present manuscript. There are several details that I go on to describe below:

Material and Methods. Lines 476-477. The authors wrote: Virgin females were dissected 3-7 days after blood-feeding, and the egg clutches were counted. If females must be dissected (as indicated), these “eggs” are not still oviposited. Please, define at which stage an egg chamber is considered an egg. Must it have the chorion completed? At which stage of development are the egg chambers of the dissected treated females? Moreover, in the introduction (lines 59-60) is stated that the reproductive cycle culminates in the development of a full set of eggs in about 2-3 days. Why the dissections were done 3-7 days after blood feeding?

Lines 139-142. More than an atypical localization it seems that Vg is not completely released from the fat body. Is it also accumulated in the hemolymph? To allow the protein to reach the oocyte the VgR must be active. Is the VgR present in the oocyte membrane? At 24h PBM, the Vg levels in ovaries seem similar (Figure S1C) in dsLp and controls. However, in the image (Figure S1D) the ovaries are less developed. Do you have any hypothesis to explain that? Later, at 48 hours PBM the egg chambers appear completely reabsorbed.

Line 146. Indicate Figure 1D after “ovaries”. In the next line indicates Figure 1C.

Line 146. Figure 1D. From this figure, it seems that follicular cells were affected by Lp depletion. Are they in apoptosis? By the way, did you observe the nurse cells? Are they also affected? It is affected the LpR in oocytes?

Line 147. Vg does not disappear from western blots. It is not detected.

Figure 1 legend. Regarding Fig. 1A, the authors say that "each dot represents eggs per female", and "N= number of females"(86 in the case of controls). However, the legend further refers to the statistics and goes saying "(Mann-Whitney: ****= p<0.0001; FOUR biological replicates)"(capitals are mine). What is finally the number of replicates? A similar confusion appears in panels B and C, and in diverse panels in Figure 2, Suppl. Figure 2 and Suppl. Figure 3.

Lines 180-182. After dsVg treatments, the depletion of Vg expression is complete (Fig S2 and Fig 2G). Can the authors explain why an important number of females can develop a similar number of “eggs” than control females? (Fig 2A). Indicate in the legend if these “eggs” are quantified from ovaries of 3-7 day PBM females, as indicated in material and methods. By the way, the interval of 3-7 days is too large in a mosquito life to describe phenotypes, perhaps the authors can provide the data corresponding to each day and the result could show a kinetic of egg chamber development after dsVg treatment.

Lines 178- 188. From the title of the section and the title of Figure 2 legend, a description of oocyte development (or egg chamber development) was expected. However, in the text and Figure, all the data refer to eggs. The explanation in material and methods (see the former comment) indicates that females were dissected to quantify the number of eggs (perhaps egg chambers). In line 181, the authors noted that a decrease in Vg led to a significant reduction in egg count. Should we assume that Figure 2A depicts oviposited eggs and Figure 2B portrays the larvae hatching from those eggs? Please, clarify these points. Indicate if these females are virgins or fecunded.

Line 183. The authors wrote: “no embryos hatching from Vg-depleted mothers” Which is the meaning of hatching in this sentence. Are you meaning hatching larvae from eggs?

Line 190. From Figure 2D, it seems that the nurse cells are bigger 24 hours after the dsVg treatment than in control females. Any hypothesis explaining that?

Line 195. Change silencing by depleting. Revise the document, the mRNA cannot be silenced by dsRNA.

Lines 195-200. After the peak at 24 hours, Lp expression reaches the control levels at 48 h PBM. Do you have any suggestions for this return to normal levels?

Line 237. Treatment with rapamycin has resulted in a 50% decrease in Vg. Have higher doses been attempted to further decrease Vg? In dsVg-treated females, there is an increase in p-6SK. Do you analyze the p-6SK in dsLp-treated females? The changes in TOR signaling only happen when Vg is depleted?

Line 241. Change “post PBM” by “PBM”

Line 243. Add (Figure 3B) after “controls”

Lines 312-327. Please revise the description in this section. It is confusing the use of “embryos” when the authors are making references to eggs. For example:

Line 314. Authors say “the mechanisms causing infertility in embryos”. Please revise, the word infertility is not correct in this sentence. Perhaps arrest of development, or egg infertility.

Line 315. Embryos cannot be fertilized. The eggs can be fertilized.

Line 322. Embryos are not deposited in mosquitos

Line. 356. Authors wrote: “(B) Lp is deposited into dsVg-derived embryos”. Lp was deposited in the eggs.

Figure 4. Please explain the figures in panel A. What are showing the two images of controls? It seems that both are at the surface of the eggs. Considered to change one of them for an image at a higher magnification that allows the comparison with the image from the treated embryo.

Line 586. It is indicated “5 2mm glass” probably it must say 5.2 mm

Through the document change ml by mL. The same, change μl by μL and nl by nL

Revise the correct use of “silence” after treatment with dsRNA. The RNAi do not silence the gene expression.

Reviewer #2: This article reveals that LP and Vg are reciprocally regulated in Anopheles gambiae, functioning as two nutrient transporters that maintain the normal development of ovaries and ensure fertility. Silencing LP disrupts lipid transport, leading to the mis-regulation of Vg expression in follicles and enlargement of yolk granules, ultimately resulting in the cessation of egg production. Conversely, silencing Vg upregulates the expression of LP in the fat body, causing excessive lipid accumulation in the follicle and rendering females completely infertile. The regulation of LP by Vg is partially mediated by TOR signals. Additionally, the study found that Vg can control amino acid levels and protein synthesis by transporting amino acids, which may influence embryonic development. In brief, the manuscript's data is detailed, and the results appear reliable, but there are some issues that need to be addressed.

Suggestions:

Major:

1. Figure 1, add the result of hatchability in Lp-silenced females.

2. Figure 1B, the authors measured the levels of triglycerides in the midgut and ovary, indicating triglycerides accumulate in the midgut and do not transfer from the midgut to the ovary. But why not test the content of triglycerides in fat body.

3. Supplemental Figure 1B (where Vg expression was measured only in fat bodies), whereas in supplemental Figure 1C fat bodies and ovaries were used. It is not comparable.

4. For Figure 2, Supplemental Figure 1 and 2, the control is actin sometimes and changes to Rpl19 in other cases.

5. Figure 4B, the disappearance of actin, even if protein synthesis is inhibited, requires explanation, as actin, a housekeeping gene, must be present in living cells. Please provide a reasonable justification for this observation.

6. Problems with lipid metabolism. In Figure 2G, Lp protein increased in both fat body and ovary of Vg-silenced females. Meanwhile, in Figure 2E, triglyceride levels were also elevated in the ovary of Vg-silenced females. However, in Figure 4B, Lp protein decreased in the embryos of Vg-silenced females and in Figure 4D, and triglyceride levels were still elevated. How to explain this contradictory result? Importantly, in Figure 4D, supplemental 4B and 4C, no differences in total lipids were observed in dsVg-derived embryos relative to controls, but triglyceride levels increased, and some lipid levels decreased. These results only prove that lipids changed in Vg-silenced females, but the relationship between lipid alterations and reproduction is not explained, so it is recommended that further experimental validation or discussion of the lipid metabolism results is needed.

Minor:

7. Please ensure the consistent use of units and correct punctuation throughout the article. For instance, there are inconsistencies in punctuation, such as line 147: “2mm” and line 283: “40 μM”.

8. Line 127-129, what does “without characterizing the phenotype” mean here?

9. Line 145-148, Whether the sample is ovaries (figure legend, line162) or eggs (result, line145).

10. Line 147, the phrase “largely disappeared” may not be the most accurate description of Figure 1C. Consider using more precise description.

11. Line 390, “Vg localization is affected (Figure 1C)” change to “Vg localization and accumulation is affected (Figure 1C)”.

12. In Figure 4B, it would be beneficial to include a biological statistical analysis to more intuitively illustrate the changes in yolk granules.

13. In Supplemental Figure 3E, I'm curious why qPCR analysis of LP and Vg expression levels was performed solely on fat bodies and not on ovaries. Please clarify.

Reviewer #3: In this paper, Stryapunina et al. describe the interplay between nutrient transporters Lipophorin (Lp) and Vitellogenin (Vg) to ensure fertility in the malaria mosquito Anopheles gambiae. The authors used a plethora of methods to describe the mutual relationship of studied factors and show effects on female mosquito physiology. The manuscript is well written, experiments are robust and well described and the story is easy to follow. Although the authors do not provide direct experimental evidence of how at a molecular level Lp and Vg mutually affect their expression and function, the work is on a track to uncover basic regulation of the gonotrophic cycle in a mosquito species, which is a deadly threat to many human populations, and thus, studies of its reproduction are of the highest interest. I have several major and minor comments on the manuscript.

Major comments

My major concern is the described relationship of Lp and Vg. Although the authors describe Lp and Vg relationship as an ‘interplay’ in the article title, there is no experimental proof, in the sense of a molecular mechanism, of a direct effect of Lp on Vg expression and Vg on Lp. Several intermediate steps between the Lp silencing and changed Vg levels, or Vg depletion and changed Lp levels could be included and drive the ‘interplay’. The authors themselves mention in the discussion that Lp expression induction upon Vg RNAi can be just a consequence of the presence of free amino acids, which were not incorporated in Vg but remained in the FB, activating the TOR signaling pathway. I need to admit that the authors are aware of what was just said and are mostly cautious about their conclusions, which is appropriate given the data they present.

1) The authors mention in the introduction, that Lp shuttles not only neutral lipids from the midgut to the ovaries but also cholesterol. It is difficult to rule out the role of ecdysone from Lp/Vg interplay with Lp as a shuttle of ecdysone precursor cholesterol and a (putative) target of the ecdysone signaling pathway. Moreover, with ecdysone importance for Vg expression. In the Suppl. Fig 1B, Vg mRNA expression in the FB at 24h PBM is higher in control dsLacZ females compared to dsLp females. Could Lp-deprived females suffer from lower ecdysone production? Could that be reflected in the lower Vg transcription?

Further, Lp seems to be repressed by ecdysone signaling in An. gambiae (Werling et al. 2019) and Lp mRNA expression peaks at 12h PBM (Fig 2F; Werling et al., 2019), ahead of the ecdysone peak (Redfern 1982; Pondeville et al., 2008). However, the maximal Lp expression in the Vg-RNAi background is shifted to 24h PBM (when ecdysone and free amino acids are present) but drops at 36h and 48h PBM, when free amino acids are present in the FB (Suppl Fig 3), but ecdysone is dropping (mainly at 48h PBM). Could ecdysone (with amino acids?) function as a feedback loop to repress Lp expression (and function) and really coordinate Lp/Vg interplay? Do TOR and 20E signaling really exert opposite effects on Lp expression as suggested in the discussion? I believe those are serious questions, that should be considered, discussed, and hopefully supported by data.

2) Vg protein level in the FB is higher in Lp-silenced females at 24h PBM. Does Lp directly block Vg release to the hemolymph? Or are ovaries already degenerating and thus not accumulating Vg even at 24h PBM and the excess of the Vg is just retained in the FB? The authors might discuss it a little more.

3) The authors consistently mention the number of biological replicates, but the number of plotted and statistically analyzed samples/animals or FB/ovary pools – the total number ‘N’ – is sometimes missing or can be confused with the number of biological replicates. The authors should add ‘N’ to the figure: Fig 1B, 2E, Suppl Fig 2B-C etc.

4) Gene symbols in many cases describing samples/animals upon dsRNA injection are often not in italics, especially dsLp and dsVg.

Minor comments:

Line 33-34 ‘Embryos deposited by Vg-depleted mothers are completely infertile…’ I think only Vg-depleted mothers are completely infertile.

Line 217 (Suppl. Fig 1D) Please remove 12h and 36h time points (x-axis), as no values are plotted for those time points.

Line 227 ‘Vg knockdown by second target …’ It is Vg, which is targeted by the second Vg RNAi fragment.

Line 257 ‘that in normal conditions Vg expression represses TOR signaling’ – Is it really repression as we commonly understand it or does Vg expression lead to repressed TOR pathway?

Line 281 The Vg-knockdown mediated.

Line 286-287 ‘in the same four groups’ might suggest that the same samples as in Fig 3D were used, which was not the case. Please reword the caption accordingly.

Line 350 (Figure 4A) It is difficult to see details in Figure 4A. Although the figure quality is supposed to be higher in the final version, the authors could change the blue channel color to white.

Line 388 Is 20E ‘synthesized in the fat body’, or is it converted from ecdysone synthetized in ovaries?

Line 424 Is it really repression as we commonly understand it? Or Vg expression leads to TOR signaling suppression.

Line 471 I am quite skeptical that the authors had dsRNA of 10 ug/uL concentration, especially if NanoDrop was used for quantification. NanoDrop measures the concentration linearly up to ~ 5 ug/uL and the higher concentration is just a linear approximation, not an accurate measurement. How confident the authors are in dsRNA concentration determination?

Line 476-477 Were females really dissected to do the egg counting?

Line 489-490 Why were random primers used for the cDNA synthesis? Were they more reliable than oligo(dT) primers?

Line 495 and Suppl. Table 4 Rpl19 and genes in the table in italics.

Line 513 What does 25% bleach mean? Please provide the final concentration of the active substance(s).

Line 537-540 Was acetone part of the solution? Wasn’t rapamycin dissolved in DMSO only?

**Have all data underlying the figures and results presented in the manuscript been provided?**

Reviewer #1: Yes

Reviewer #2: Yes

Reviewer #3: **No: **The authors should provide numerical data that underlies graphs in spreadsheet form as supporting information.

PLOS authors have the option to publish the peer review history of their article (what does this mean?). If published, this will include your full peer review and any attached files.

Reviewer #1: No

Reviewer #2: **Yes: **zhen zou

Reviewer #3: No

---

## [Decision Letter · Decision Letter 1]

28 Dec 2023

Dear Dr Catteruccia,

Thank you very much for submitting your Research Article entitled 'Interplay between nutrient transporters ensures fertility in the malaria mosquito *Anopheles gambiae*' to PLOS Genetics.

The manuscript was fully evaluated at the editorial level and by independent peer reviewers. The reviewers appreciated the attention to an important topic but identified some concerns that we ask you address in a revised manuscript.

We therefore ask you to modify the manuscript according to the review recommendations. Your revisions should address the specific points made by each reviewer.

Yours sincerely,

Takaaki Daimon

Academic Editor

PLOS Genetics

Gregory P. Copenhaver

Editor-in-Chief

PLOS Genetics

Reviewer's Responses to Questions

**Comments to the Authors:**

Associate Editor: Thank you for revising the manuscript. The manuscript is almost ready for publication, but as you will see below, Reviewer #3 has some minor comments. I'd like to ask you to address these comments in the final round of review.

Reviewer #1: My comments have been addressed, and the manuscript fulfills my expectations.

Reviewer #2: I support the publication of this manuscript.

Reviewer #3: In the manuscript by Stryapunina et al., the authors present a revised version of the original submission. I appreciate the authors for their response to the suggestions from all the reviewers and for providing new data. The changes incorporated into the manuscript substantially improved its quality, accuracy, and readability. The manuscript is ready to be accepted after dealing with only several small issues.

I still think the authors should reconsider changing the title to reflect more the fact that the Lp/Vg interaction is most likely indirect, and the term interplay is not fully correct, from my point of view, interplay suggests either a physical interaction or direct involvement in the other’s protein actions. I understand that the term ‘interplay’ is “catchier”, but I would suggest something like (examples) Orchestration of nutrient transporters ensures fertility in the malaria mosquito Anopheles gambiae OR Precise coordination between nutrient transporters ensures fertility in the malaria mosquito Anopheles gambiae.

The fact that the cholesterol from the larval stage seems to be used for ecdysteroidogenesis is interesting. It might safeguard the vitellogenic females from being dependent on blood meal-originated cholesterol and enable them to promptly start ecdysteroidogenesis and the first gonadotrophic cycle, without losing time for Lp to transport cholesterol from the gut.

Lines 118-119 This is tightly linked to what I have already mentioned above. Do the authors expect the Vg to be the molecule, which is directly responsible for the termination of the Lp-mediated accumulation of lipids in the ovaries? Or is Vg a proxy of lacking amino acids?

Lines 152 might suggest the Vg mRNA is transcribed in the FB and Vg mRNA is incorporated into the developing oocyte. The authors could change it to (example) ‘…the major nutrient transporter, which as mentioned above, is transcribed and translated in the fat body and incorporated into developing oocytes.’

Line 273 The authors say: ‘and is in turn regulated by this yolk protein’ Is the yolk protein Vg or amino acids not incorporated into Vg?

Lines 322-326 and S4D Fig I think the authors should add triglyceride levels presented in Fig 4D to the S4D Fig for better comparison with other lipids.

Lines 347-348 add reference: ovaries are degenerating and cannot accumulate further yolk (S1D Fig)

Line 351 word ‘overload’ might be too strong given the fact the level of triglycerides is about 1.2x-1.3x higher in Vg-depleted females compared to controls.

Finally, I wish you a Happy Holidays and a peaceful and prosperous New Year!

**Have all data underlying the figures and results presented in the manuscript been provided?**

Reviewer #1: Yes

Reviewer #2: Yes

Reviewer #3: Yes

PLOS authors have the option to publish the peer review history of their article (what does this mean?). If published, this will include your full peer review and any attached files.

Reviewer #1: No

Reviewer #2: No

Reviewer #3: No

---

## [Editor Report · Decision Letter 2]

20 Jan 2024

Dear Dr Catteruccia,

We are pleased to inform you that your manuscript entitled "Precise coordination between nutrient transporters ensures fertility in the malaria mosquito *Anopheles gambiae*" has been editorially accepted for publication in PLOS Genetics. Congratulations!

Yours sincerely,

Takaaki Daimon

Academic Editor

PLOS Genetics

Gregory P. Copenhaver

Editor-in-Chief

PLOS Genetics

Comments from the reviewers (if applicable):

Thank you for revising the manuscript.

I think it is now ready for publication.

**Data Deposition**

http://datadryad.org/submit?journalID=pgenetics&manu=PGENETICS-D-23-00835R2

**Press Queries**

---

## [Editor Report · Acceptance letter]

25 Jan 2024

PGENETICS-D-23-00835R2 

Precise coordination between nutrient transporters ensures fertility in the malaria mosquito *Anopheles gambiae*

Dear Dr Catteruccia, 

We are pleased to inform you that your manuscript entitled "Precise coordination between nutrient transporters ensures fertility in the malaria mosquito *Anopheles gambiae*" has been formally accepted for publication in PLOS Genetics! Your manuscript is now with our production department and you will be notified of the publication date in due course.

With kind regards,

Bernadett Koltai

PLOS Genetics

On behalf of:
